# Resistance Breeding for Northern Corn Leaf Blight with Dominant Genes, Polygene, and Their Combinations—Effects on Disease Traits

Xiaoyang Zhu *, Lana M. Reid, Tsegaye Woldemariam, Jinhe Wu, Krishan K. Jindal and Aida Kebede

Ottawa Research and Developmental Center, Agriculture and Agri-Food Canada, Ottawa, ON K1A 0C6, Canada; hcd69@hotmail.com (L.M.R.); tsegaye.woldemariam@agr.gc.ca (T.W.); jinhe.wu@agr.gc.ca (J.W.); jindal.krishan@gmail.com (K.K.J.); aida.kebede@agr.gc.ca (A.K.)
* Correspondence: xiaoyang.zhu@agr.gc.ca

**Abstract:** Resistance breeding is the most effective method to control northern corn leaf blight (NCLB). The objectives were to (1) Assess effects of dominant genes (*Ht*(s)), polygene (PG), and their combinations to disease rating (DR), number of lesions per leaf (NLPL), and lesion size (LS); (2) Estimate genetic components, general combining abilities (GCA), and heritability under two Line × Tester analyses; and (3) Determine gene action through mid-parent heterosis (MPH) and better-parent heterobeltiosis (BPH) analysis. A total of 163 genotypes, including 120 crosses, their parents, and 10 hybrid checks, were evaluated under two NCLB artificial inoculations in 2015 and 2016. The results indicated that PG had the best resistance to DR, NLPL, and LS in crosses, followed by PG*Ht*(s) and single *Ht*(s). *Ht1* had both resistant and susceptible lesions. *Ht2* and *Ht3* expressed more resistance to LS significantly, while *Htm1* and *Htn1* had more resistance to NLPL. *Htm1*/*Ht2*, PG/*Htm1*, and the other 11 combinations were found excellent for NCLB resistance. Line × Tester analysis showed that additive effects were more important, and GCA of *Ht*(s) reduced disease traits. However, lower narrow sense heritability indicated that additive effects were low. MPH and BPH results showed that dominant and over-dominant gene actions existed for DR and NLPL.

**Keywords:** corn; disease rating; heritability; heterosis; heterobeltiosis; lesion size; Line × Tester; northern corn leaf blight; number of lesions per leaf; resistance genes; *Ht1*; *Ht2*; *Ht3*; *Htm1*; *Htn1*; polygene

## 1. Introduction

Northern corn leaf blight (NCLB), caused by *Exserohilum turcicum* (Passerini) Leonard and Suggs = *Setosphaeria turcica* [Luttrell] Leonard and Suggs = *Helminthosporium turcicum*, is the most common and economically significant leaf disease of corn (*Zea mays* L.) worldwide. NCLB occurs during relatively cool and wet seasons [1]. Sporulation requires a 14 h dew period and a temperature of 20–25 °C [2]. Periods of wetness that last more than 6 h at a temperature of 18–27 °C are most conducive to NCLB infection [3]. Under favorable conditions, susceptible lesions can form in 12 days [1,4]; thus, one complete disease cycle on susceptible plants takes place within 14 days, whereas it takes about 20 days on resistant plants [2,3,5]. Typically, NCLB causes 15–30% yield losses [6]. Heavy infections of NCLB can cause grain yield losses ranging from 40% [7,8] up to 70% [6] and silage losses up to 91% [9]. In Canada, NCLB became an economically important foliar disease recently. It infected 98% of surveyed fields in 2010 in Ontario and Quebec [10] and 97% of the field in 2015 in Ontario [11], and yield losses could reach up to 50% [12].

Resistance breeding is the most economical, sustainable, and effective way to control NCLB. In NCLB studies, resistance expression included lesion type [13–16], disease score = scale = rating [17,18], area under the disease progress curve (AUDPC), apparent infection rate (r) [8,19], yield loss, kernel weight loss [8], number of lesions per plant, lesion size, area of the sporulating zone, number of conidia produced on lesion segments [20],

and percent leaf area affected [21]. AUDPC and latent period were used for the partial resistance study [22].

Resistant sources of NCLB and their resistant genes were identified in corn breeding history. In 1956, 1066 accessions of corn germplasm were evaluated for resistance to NCLB [23]. Eighty-four sources had chlorotic lesion resistance [18]. There are 3286 accessions with NCLB resistance results recorded in U.S. National Plant Germplasm System (https://npgsweb.ars-grin.gov/gringlobal/descriptors.aspx, (accessed on 8 March 2023), including 991 accessions with high to average resistance (rating 1–4) in 2022. Welz and Geiger [24] reviewed all genes for resistance to NCLB in a diverse maize population. The dominant gene *Ht1* was found in inbred GE440 and Ladyfinger popcorn [13–16]; *Ht2* was found in NN14 [25]; *Ht3* was incorporated from *Tripsacum floridanum* into corn [26]; *HtN = Htn1* was derived from Pepitilla [27]; *Htm1* was derived from Mayorbela [28]; *HtP* was derived from an American resistant line L10 [29,30]; *HtNB* was derived from an Indonesian variety Bramadi [31]; and two recessive resistant genes, *ht4* and *rt*, were found in synthetic BS19 [32] and L40 [29,30], respectively. The dominant gene *NN* from Thailand-inbred Ki14 was also mentioned by Welz and Geiger [24]. In recent Genome-wide association studies, quantitative trait locus (QTL) for resistant genes were detected from different populations [33–39].

*Ht1*, *Ht2*, *Ht3*, and *Htn1* were used for physiological race identification. Based on the new system for the designation of physiological races [40], races 0, 1, 2, 23, 2 N, and 23 N existed in the United States [41–47]; 0, 1, 3, N, 12, 13, 23, 1 N, 2 N, 3 N, 12 N, 23 N, and 123 in China [48]; 23, 2 N, and 23 N in Mexico; 0, 23, and 23 N in Zambia, 0, 1, 2, 3, N, and 23 N in Uganda [49]; 0, 1 N, 12 N, and 123 N in Brazil [50]; 0 and 1 in Serbia [51]; and 0, 1, 2, 3, N, 12, 13, 23, 3 N, 123, 13 N, and 23 N in Kenya, Germany, and Austria [52]. *Ht1*, *Ht2*, *Ht3*, *Htm1*, and *Htn1* were used for physiological race identification recently. Twenty races (0, 1, 2, M, N, 12, 13, 23, 1 M, 1 N, MN, 123, 1 MN, 2 MN, 23 M, 23 N, 12 MN, 23 MN, 123 M, and 123 MN) existed in north central United States [53]; seventeen races (0, 1, 2, 3, M, N, 12, 1 M, 1 N, 3 M, 13 M, 12 N, 13 N, 1 MN, 12 MN, 13 MN, and 123 MN) were found in Ontario, Canada [54].

*Ht1* was introduced into 30 public inbreds [17] and was used extensively during the late 1960s and 1970s [32] and later [42,55]. Inbred Oh43, Oh45, and Mo17 had polygenic resistance and were combined with monogenic *Ht1*, *Ht2*, and *Htn1* for resistance study; polygenic combined with *Htn1* had better resistance than with *Ht2* and *Ht1*, and monogenic combination *Ht2/Ht1* had better resistance than *Ht2/Ht2* and *Ht1/Ht1* [20]. In a study inoculated with race 0 and 1, *Ht1* + partial resistance hybrid sweet corn had the best resistance, followed by partial resistance, *Ht1*, and susceptible hybrids [56]. Similar results were found in grain corn hybrids crossed with partial resistance line H99 [57]. H99 was estimated to have 2–3 genes for lesion number and 4–7 genes for the percent leaf area affected [21].

The traditional resistance breeding methods include population improvement and backcross. If polygene resistance is the only source available, population improvement should be used to make a more resistant population. When the dominant resistant gene is available, the backcross method can incorporate a resistant gene from the donor to have higher yield materials or to have materials with polygene resistance or partial resistance for better resistance. From 2006 to 2014, based on the backcross method, *Htm1* and *Htn1* were successfully introduced into a susceptible inbred CO388, and *Ht1*, *Ht2*, *Ht3*, *Htm1*, and *Htn1* were successfully introduced into a polygenic resistant inbred CO428. The objectives of this paper were to: (1) estimate single and polygene effects and better gene combinations to NCLB resistance, such as disease rating (DR), number of lesions per leaf (NLPL), and lesion size (LS); (2) estimate genetic components, general combining ability (GCA), and specific combining ability (SCA), heritability (H), and the narrow sense heritability (H(ns)) by Line × Tester analysis; and (3) estimate gene action under different genotype backgrounds, additive (A) and dominant (D) gene effects through mid-parent heterosis (MPH), and better-parent heterobeltiosis (BPH) analysis.

## 2. Materials and Methods

### 2.1. Breeding History

Based on previous corn germplasm introduced from the United States (USDA-MWA-PIRU and NCRPIS, 1305 State Avenue, Ames, IOWA 50014, United States; and Cornell University, United States), seed increasing and disease screening to NCLB resulted [58,59] in 7 accessions with *HT* = *Ht1* gene (Ames 23458, A619*HT*; Ames 23468, A632*HT*; Ames 27065, B73*Htrhm*; Ames 27138, N28*HT*; PI 600729, LP1NR*HT*; PI 600755, LP1 CMS *HT*; and, PI 601079, LH123*HT*), 2 accessions with *Ht1* gene (Ames 25219, A619*Ht1* and Ames 25372, Pa91*Ht1*), 2 accessions with *Ht2* gene (Ames 25220, A619*Ht2* and Ames 25373, Pa91*Ht2*), 2 accessions with *Ht3* gene (Ames 25221, A619*Ht3* and Ames 25374, Pa91*Ht3*), 2 accessions with *Htm1* gene (PI 550496, H102) [28] and inbred 73,353 [60], and 6 accessions with *Htn1* gene (PI 406112, A214*N*; PI 406118, A509*N*; PI 406119, A553*N* (Orange Halo); PI 406120, A553*N* (Red Halo), PI 406126, A661*N*; and, Ames 23469, A632*HtN*). In total, 21 resistant sources were planted and crossed with two inbred lines, CO388 and CO442, with high general combining ability (GCA) for grain yield and lower GCA CO428 but with polygene (PG) resistance to NCLB, gray leaf spot, eyespot, Stewart's wilt, and Goss's wilt [58]. Their pedigree and heterotic groups are listed in Table 1. Most resistance sources come from the non-stiff stalk (NS) group, Oh43 subgroup, and tropical materials. However, A632*HtN* and B73*Htrhm* belong to the stiff stalk (SS) group, B14 and B73 subgroups, respectively. The brief breeding histories were: 20 CO388 × R sources, 12 CO442 × R sources, and 15 CO428 × R sources crosses made in 2006; In 2007, all crosses were artificially inoculated twice [61], based on their resistance, both lesion type and DR; all susceptible crosses which only showed susceptible lesion or DR > 5.5 at the late stage were deleted, including all CO442 crosses. Only 9 CO388 × R sources and 9 CO428 × R sources were backcrossed with CO388 or CO428, respectively. The same inoculation and selection methods were used in 2008; 3 families of (CO388 × R sources)CO388 were deleted. Only 6 families of (CO388 × R sources)CO388 and all 9 families of (CO428 × R sources)CO428 were backcrossed with CO388 or CO428, respectively. Similarly, in 2009, 3 families of (CO388 × R sources)CO388(2) were deleted. Only 3 families of (CO388 × R sources)CO388(2) and all 9 families of (CO428 × R sources)CO428(2) were backcrossed with CO388 or CO428, respectively. In 2010, all families were backcrossed. Three families of (CO388 × R Sources)CO388(4) were simplified and named BLT01 to BLT03, and 9 families of (CO428 × R sources)CO428(4) were named BLT05 to BLT13. These families were ear-row selfed in 2011, 2012, and 2013 in Ottawa and 2011 winter in New Zealand. BLT08 was deleted in 2013 because the inbred yield was not as well as CO428. From 2013 to 2015, these families were crossed with tester lines. During 2014 to 2016, selfed lines were re-screened with artificial inoculation, and yield trials were tested.

**Table 1.** Accession #/Sources/Pedigrees, heterotic groups, assumed resistant genes, and days to silking of 35 inbreds and their disease ratings, and lesion types under an artificial epidemic of northern corn leaf blight.

| Name/Code | Purpose | Accession #/ Source/Pedigree | Heterotic Group | Assumed Resistant Gene | Days to Silking | Disease Rating | Lesion Type |
|---|---|---|---|---|---|---|---|
| A619 | Tester | PI 587139, USA | NS-Oh43 | None | 83 | 6.6 | S |
| A619*Ht1* | Tester | Ames 25219, USA | NS-Oh43 | *Ht1Ht1* | 84 | 5.0 | R/S |
| A619*Ht2* | Tester & R source | Ames 25220, USA | NS-Oh43 | *Ht2Ht2* | 84 | 4.7 | MR/MS |
| A619*Ht3* | Tester | Ames 25221, USA | NS-Oh43 | *Ht3Ht3* | 85 | 4.9 | MR/MS |
| A632*HTN* | R Source | Ames 23469, USA | SS-B14 | *Htn1Htn1* | 85 | 5.2 | MR/MS |
| A509*N* | R Source | PI 406118, USA | Tropical | *Htn1Htn1* | 92 | 5.0 | R/MR |
| A553*N* | R Source | PI 406119, USA | Tropical | *Htn1Htn1* | 92 | 3.6 | R/MR |

**Table 1.** *Cont.*

| Name/Code | Purpose | Accession #/ Source/Pedigree | Heterotic Group | Assumed Resistant Gene | Days to Silking | Disease Rating | Lesion Type |
|---|---|---|---|---|---|---|---|
| 73353 | R Source | Cornell University, USA | Tropical | *Htm1Htm1* | 87 | 3.2 | R/MR |
| H102 | R Source | PI 550496, USA | Tropical | *Htm1Htm1* | 89 | 2.9 | MS |
| LH123*HT* | R Source | PI 601079, USA | NS-Oh43 | *Ht1Ht1* | 96 | 4.0 | R/S |
| Pa91 | Inbred check | PI 587147, USA | SS | None | 91 | 4.9 | S |
| Pa91*Ht1* | Inbred check | Ames 25372, USA | SS | *Ht1Ht1* | 92 | 3.9 | R/S |
| Pa91*Ht2* | R Source | Ames 25373, USA | SS | *Ht2Ht2* | 94 | 3.6 | MR/MS |
| Pa91*Ht3* | R Source | Ames 25374, USA | SS | *Ht3Ht3* | 91 | 3.2 | MR/MS |
| CO353 | Inbred check | AAFC | SS | Unknown | 87 | 3.9 | MR/MS |
| CO388 | Line × tester | AAFC, (B73 × CO272) CO272 | SS-B73 | None | 82 | 6.5 | S |
| BLT01 | Line × tester | AAFC, (CO388 × A553N)CO388(4) | SS-B73 | *Htn1Htn1* | 85 | 5.7 | MR/MS |
| BLT02 | Line × tester | AAFC, (CO388 × A632HtN)CO388(4) | SS-B73 | *Htn1Htn1* | 81 | 5.4 | MR/MS |
| BLT03 | Line × tester | AAFC, (CO388 × H102)CO388(4) | SS-B73 | *Htm1Htm1* | 83 | 5.6 | MR/MS |
| CO428 | Line | AAFC, OH43 × H99 | NS-Oh43 | PGPG | 84 | 3.1 | MS |
| BLT05 | Line | AAFC, (CO428 × 73353)CO428(4) | NS-Oh43 | PG*Htm1* PG*Htm1* | 81 | 3.5 | MR/MS |
| BLT06 | Line | AAFC, (CO428 × A509N)CO428(4) | NS-Oh43 | PG*Htn1* PG*Htn1* | 85 | 2.8 | R/MR |
| BLT07 | Line | AAFC, (CO428 × A619Ht2)CO428(4) | NS-Oh43 | PG*Ht2* PG*Ht2* | 83 | 3.5 | MR/MS |
| BLT09 | Line | AAFC, (CO428 × A632HtN)CO428(4) | NS-Oh43 | PG*Htn1* PG*Htn1* | 85 | 2.9 | R/MR |
| BLT10 | Line | AAFC, (CO428 × H102)CO428(4) | NS-Oh43 | PG*Htm1* PG*Htm1* | 83 | 2.2 | MR/MS |
| BLT11 | Line | AAFC, (CO428 × LH123Ht)CO428(4) | NS-Oh43 | PG*Ht1* PG*Ht1* | 84 | 2.5 | R/MR |
| BLT12 | Line | AAFC, (CO428 × Pa91Ht2)CO428(4) | NS-Oh43 | PG*Ht2* PG*Ht2* | 85 | 1.7 | R/MR |
| BLT13 | Line | AAFC, (CO428 × Pa91Ht3)CO428(4) | NS-Oh43 | PG*Ht3* PG*Ht3* | 85 | 1.8 | R/MR |
| CL30 | Tester | AAFC | NS-Flint | None | 70 | 7.0 | S |
| CO442 | Tester | AAFC | NS-Iodent | None | 79 | 7.0 | S |
| T1 | Tester | Thurston Genetics, USA | NS-Iodent | None | 82 | 7.0 | S |
| T2 | Tester | Thurston Genetics, USA | NS-Iodent | None | 79 | 5.8 | SMS |
| T3 | Tester | Thurston Genetics, USA | NS-Iodent | Partial | 78 | 6.1 | S |
| T4 | Tester | Thurston Genetics, USA | SS-B14 | None | 78 | 6.8 | S |
| T5 | Tester | MBS Genetics, USA | SS-B14 | None | 79 | 6.9 | S |

PI or Ames number is the number from USDA-MWA-PIRU and NCRPIS, 1305 State Avenue, Ames, Iowa 50014, United States. AAFC means Agriculture and Agri-Food Canada, Ottawa Research and Developmental Centre, Ottawa, Ontario, K1A 0C6, Canada. SS = stiff stalk group, NS = non-stiff stalk group; None = no resistant gene, PG = polygene resistance, Partial = partial resistance. A509*N* and LH123*HT* data from previous records.

## 2.2. Experiment Method

Based on the seeds available, a total of 163 genotypes, including 33 inbred lines and 120 crosses, and 10 commercial hybrid checks, were used for NCLB artificial inoculation in 2015 and 2016. Except for A509*N* and LH123*HT* in Table 1, all other 33 lines were included in this Study. Stiff stalk (SS) line CO388 and its *Ht* version BLT01, BLT02, and BLT03 were used as lines to cross with a non-stiff stalk (NS) A619, A619*Ht1*, A619*Ht2*, A619*Ht3* (Oh43 group), CL30 (early flint group), CO442, T1, T2, and T3 (Iodent group); these were also used as testers to cross with NS-Oh43 lines CO428 and its *Ht* version BLT05-BLT13. CO428 and its *Ht* version BLT05-BLT13 also crossed with CL30, CO442, and two SS-B14 group lines, T4 and T5. To evaluate *Ht* gene effects for NCLB development, CO428 and its *Ht* version BLT05-BLT13 also crossed with the same group A619 and its *Ht* versions. In the ANOVA statistics, CO388 was only used as a tester; therefore, it had 11 lines, 14 testers, and 8 inbred checks. Ten commercial hybrids were used as hybrid checks, including 3 hybrids with the *Ht* gene. A randomized complete block has 3 replications, a one-row plot with a row distance of 0.76 m and a 3.5 m row length for 20 plants. To reduce the shade effects of hybrid to inbred, this randomized complete block was modified by separating inbreds and hybrids into two parts in each block by adding one row inbred; inbreds and hybrids were randomized in their parts. All genotypes were planted on May 7 at Centre Experimental

Farm, Ottawa Research and Development Centre, Agriculture and Agri-Food Canada in both 2015 and 2016.

Two artificial inoculations were used to create an epidemic environment. The first inoculation time was at the 6–8 leaf stage on June 18 in both 2015 and 2016; the second inoculation time was at the 10–12 leaf stage on July 2 and July 4 for 2015 and 2016, respectively. Collected infected leaves from previous NCLB nursery and yearly corn disease survey in Ontario, Canada were ground to powder as inoculum, and two doses of powder (equivalent to 0.2 g) from a Bazooka (Sistrunk Inoculators, Starkville, MS, USA) were injected into the whorl of each plant [61]. To make an environment that favors disease development and epidemic, 10–15 min of irrigation (equivalent to 5–8 mm rainfall) from above the plants was conducted in the afternoon to add soil moisture and reduce air temperature from the first inoculation date except for rainy days. The irrigation must make the whorl have water in the morning, which is a key factor for a successful inoculation with leaf powder because only wet powder could produce spores for disease infection.

Specific resistances, such as the lesion type, were recorded twice, three weeks after the first inoculation and four weeks after the second inoculation. There are four lesion types: resistant lesion (R), stripe or narrow elliptical green-yellowish lesion; moderately resistant lesion (MR), narrow, long lesion with green-yellowish or -purplish margin and small elliptical gray center; moderately susceptible lesion (MS), long, elliptical, gray lesion with green-yellowish or -purplish margin; and susceptible lesion (S), long, elliptical, gray or tan colored lesion. General resistances and the disease ratings (DR) were recorded twice, in late August for early flowering genotypes or in middle September for late flowering genotypes, about 4 weeks post silk emergence. General resistance has 7 DRs where: 1 = no symptoms; $2 \leq 1\%$; 3 = 1–10%; 4 = 11–25% of leaves symptomatic; $5 \geq 50\%$ of the lower leaves are symptomatic and <25% of middle and upper leaves are symptomatic; 6 = bottom leaves are dead, >50% of the middle leaves, and <25% of upper leaves are symptomatic; and, 7 = plant is dead. Middle leaves refer to the four leaves near the primary ear emergence. If DR were uniform, then the DR was recorded as row base; If more than one rating scale in a genotype, the numbers of plants with different rating scales were counted. The average DR was used for further analysis.

In early September, the number of lesions was counted from the leaf above the primary ear, 5 plants for each row (genotype), and the average number of lesions per leaf (NLPL) was used for further analysis. Ten typical, single, developed lesions were measured in their lesion length (cm) and lesion width (cm) for each genotype. For genotypes with more than one kind of lesion type, each type measured up to 10 lesions depending on the number found. The average lesion length and lesion width were used to calculate lesion size (LS). LS (cm$^2$) = 0.75 × Lesion length (cm) × Lesion width (cm). Lesion number count and lesion size measurement were labor-intensive. To optimize efficiency, each replication was performed on the same day by the same person.

Average NLPL, LS, and DR were used for further analysis, representing three resistances to invasion, extension, and explosion (epidemics) of disease development.

### 2.3. Statistical Methods for Gene Effect Comparison

Analysis of Variance (ANOVA) using all genotypes for multi-environment trial analysis was based on the books of Crossa [62] and Gomez and Gomez [63].

Resistant genes were grouped differently to test the effects between the two groups by combining data from 2015 to 2016. Because of the sample size difference, a different comparison method was used to test the significance of the gene effect:

$$t = \frac{m1 - m2}{\sqrt{\frac{s1^2}{n1} + \frac{s2^2}{n2}}} = \frac{m1 - m2}{\sqrt{A + B}} \tag{1}$$

$$df = \frac{(A + B)^2}{\frac{A^2}{(n1-1)} + \frac{B^2}{(n2-1)}} \tag{2}$$

Formula (1) and (2) are Welch's version [64,65] of the two-sample *t*-test; it differs from Student's original version. Where m1 and m2 are the two sample means from samples of size n1 and n2, and s1 and s2 are their estimates of the standard deviation. In this case, m1 presents the group means with different resistant genes, and m2 presents the check mean, m1–m2 present the gene effect in the same group; n1 ≠ n2 = genotype number × replication number × year number = 6 × genotype number.

### 2.4. Statistical Methods for Line × Tester Analysis

To understand the genetic effects of different *Ht* genes with different backgrounds, there were two Line × Tester (L × T) experiments, CO388 and its *Ht* version lines BLT01, BLT02, and BLT03 crossed with 9 testers, and CO428 and its *Ht* version lines BLT05, BLT06, BLT07, BLT10, and BLT12 crossed with 7 testers. Two L × T experiments were analyzed separately due to different resistant backgrounds, but parent data of A619, A619Ht1, A619Ht2, and A619Ht3 were used in both L × T analysis, parent data of CO388, BLT02, and BLT03 were also used as Testers' data in CO428 related L × T analysis. The 2015 and 2016 single-year data analysis for L × T was based on the methods of Abu Assar [66] and Sharma [67]. The Analysis of Variance for two-year combined data was based on the SAS codes of Ejigu [68]. Other statistical analyses, such as GCA and SCA, the standard error for combining ability effects, and genetic components, were similar, but the mean square of pooled error of two-year results was used as $M_e$, and Year# (y = 2) × Replication# (r = 3) = 6 was used to replace r.

Genetic components formulas were as follows:

$$\sigma^2_{gca} \text{ (line)} = \text{Cov. half sib (line)} = (M_l - M_{l \times t})/(yrt) \tag{3}$$

$$\sigma^2_{gca} \text{ (tester)} = \text{Cov. half sib (tester)} = (M_t - M_{l \times t})/(yrl) \tag{4}$$

$$\text{for corn, } \sigma^2_{gca} = \text{Cov. half sib (average)} = \{[(l-1)M_l + (t-1)M_t]/(l+t-2) - M_{l \times t}\}/[yr(2lt-1-t)] = \sigma^2_A/4 \tag{5}$$

$$\text{for corn, } \sigma^2_{sca} = (M_{l \times t} - M_e)/(yr) = \sigma^2_D \tag{6}$$

Standard error (S.E.) calculation formulas for mean effects for GCA and SCA were as follows:

$$\text{S.E.(GCA for lines)} = [M_e/(yrt)]^{1/2} \tag{7}$$

$$\text{S.E.(GCA for testers)} = [M_e/(yrl)]^{1/2} \tag{8}$$

$$\text{S.E.(SCA effects)} = [M_e/(yr)]^{1/2} \tag{9}$$

$$\text{S.E.}(g_i - g_j)\text{line} = [2M_e/(yrt)]^{1/2} \tag{10}$$

$$\text{S.E.}(g_i - g_j)\text{tester} = [2M_e/(yrl)]^{1/2} \tag{11}$$

$$\text{S.E.}(S_{ij}) = [(l-1)(t-1)M_e/(yrlt)]^{1/2} \tag{12}$$

$$\text{S.E.}(S_{ij} - S_{ik}) = [(2Me/(yrl)]^{1/2} \tag{13}$$

$$\text{S.E.}(S_{ik} - S_{jk}) = [(2Me/(yrt))]^{1/2} \tag{14}$$

$$\text{S.E.}(S_{ij} - S_{kl}) = [2M_e/(yr)]^{1/2} \tag{15}$$

Formulas for heritability (H) based on genotype basis and heritability in the narrow sense (H(ns)) based on additive genetic variance were the followings:

$$\sigma^2_e = M_e \tag{16}$$

$$\sigma^2_{GY} = (M_{GY} - M_e)/r \tag{17}$$

$$\sigma^2_{G} = (M_G - M_{GY})/(ry) \tag{18}$$

$$\text{for corn, } \sigma^2_A = 4\sigma^2_{gca} \tag{19}$$

$$H = \sigma^2_G/[\sigma^2_G + \sigma^2_{GY}/y + \sigma^2_e/(ry)] = (M_G - M_{GY})/M_G \tag{20}$$

$$H(ns) = \sigma^2_A/[\sigma^2_G + \sigma^2_{GY}/y + \sigma^2_e/(ry)] = \sigma^2_A/M_G \tag{21}$$

where $M_l$, $M_t$, $M_{l\times t}$, $M_e$, $M_{GY}$, and $M_G$ are mean squares for lines, testers, L × T, and pooled error, Genotypes × Years, and Genotypes, respectively; y, r, l, and t are the number of years, replications, lines, and testers, respectively. $\sigma^2_A$ and $\sigma^2_D$ are additive and dominance variances.

*2.5. Statistical Methods for MPH and BPH Analysis*

The mean of parents and F1 hybrids from combined 2015 and 2016 data of two L × T were utilized for the estimation of heterosis. The mid-parent heterosis (MPH) and the better-parent heterobeltiosis (BPH) were estimated as follows [69]:

$$MPH(\%) = [(F1 - MP)/MP] \times 100 \tag{22}$$

$$BPH(\%) = [(F1 - BP)/BP] \times 100 \tag{23}$$

The significance of heterosis was tested using the formula as suggested by Wynne et al. [70]

$$\text{'t' over mid-parent heterosis, } t = (F1 - MP)/[2M_e/(yr)]^{1/2}, \text{ with degree of freedom of } M_e. \tag{24}$$

$$\text{'t' over heterobeltiosis, } t = (F1 - BP)/[2M_e/(yr)]^{1/2}, \text{ with degree of freedom of } M_e. \tag{25}$$

where F1 is the mean of the F1 hybrid; BP is the mean of the better parent for resistance; MP is the mean of the two parents; $M_e$ is the mean square of pooled error; y and r are the numbers of years and replications, respectively.

To explain differences between 2015 and 2016, data for climatic factors, such as daily maximum and minimum temperatures (Tx and Tn) and daily rainfall, were collected from the weather station on the farm about 200–250 m far from the experimental field. Tx and Tn are used to calculate daily corn heat units (DCHU), DCHU = [1.8 × (Tn − 4.4) + 3.33 × (Tx − 10) + 0.084 × (Tx − 10)$^2$]/2. If Tx > 30 °C, let Tx = 30. Accumulated CHU (ACHU) is the total DCHU from the planting date to the end date; if DCHU < 0, let DCHU = 0 [71]. The accumulated rainfall is the total rainfall from the planting date to the end date. It is possible to see the climatic differences between ACHU and accumulated rainfall between 2015 and 2016.

All statistics were conducted with Excel 2016 (Microsoft office Professional Plus 2016). However, R (R version 3.4.4) results from package "agricolae" [72] and SAS (SAS 9.4, SAS Institute Inc., Cary, NC, USA) results were used to make sure all Excel results were right for ANOVA and L × T analysis. Only Excel results were used because they were more accurate due to more decimals involved.

## 3. Results and Discussions

### 3.1. Inbred Data

In a cold, short-season area such as Ottawa, Canada (2700 CHU ≈ 90 days to relative maturity), four key factors were used to judge whether *Ht* genes were successfully back-crossed. (1) To make sure the dominant gene was introgressed. For NCLB, qualitative traits lesion type and margin color and quantitative trait DR were used to judge all new lines. These lines, except BLT10, had similar lesion types and the same margin colors as the donor had (Table 1). Line 73,353 and A509*N* had yellow-purplish margin, and A553*N*, A619*Ht2*,

A632*HtN*, LH123*Ht*, Pa91*Ht2*, and Pa91*Ht3* had purplish margin. H102 was without a typical margin, but when crossed with CO388, their resistant selections had a yellow-purplish margin. The purplish margin could be used as a marker to select resistant plants. (2) Their disease ratings at the late stage should be better than susceptible receptor CO388 or close to the polygene receptor CO428. For BLT10, because CO428 and H102 had similar lesions without margin color, BLT10 had a better rating (2.2) than CO428 (3.1). Table 1 shows their success. (3) Their silking or pollen-shedding date was closer to the resistant gene acceptor. Because all donors were silking 1–10 days later than CO388 and CO428, and selected lines BLT01-BLT03 were silking −1 to 3 days differently with CO388, BLT05-BLT13 were silking −3 to 1 days differently with CO428 (Table 1). (4) Selected lines kept the similar general combining ability to yield. Our results showed that under non-NCLB epidemic conditions, new lines and their crosses had similar yields; after artificially inoculating once or twice, resistant crosses had about a 4–56% yield increase. The resistant gene effects on yield traits will be reported in another paper.

All *Ht1* resistant-sources had both R and S, and other resistant sources had R and MR or MR and MS lesion types (Table 1). When *Htn1* and *Htm1* were introduced into CO388, BLT01, BLT02, and BLT03, they had the same lesion types as their resistant sources, but when *Ht2*, *Ht3*, *Htm1*, and *Htn1* were introduced into the polygene line CO428, their lesion types were better, and more R and MR lesion types appeared. Some resistant sources, such as A632*HtN*, Pa91*Ht1*, Pa91*Ht2*, Pa91*Ht3* and CO353, had purple margins. It proved three things: (1). Mixed NCLB-diseased leaf powder could be used as inoculum, which included multiple physiological races [54]. (2). Lesion type + margin color, yellow or purple, can be used similarly as a gene marker for NCLB-resistant breeding, but margin color was related to different genotype backgrounds. (3). With a PG resistance background, *Ht2*, *Ht3*, *Htm1*, and *Htn1* could improve their lesion types, which means the specific resistant lesion type can be modified by other genes.

*3.2. ANOVA Results*

The ANOVA results for DR, NLPL, and LS, by separate year, showed that most sources were significant, but replications and hybrids vs. inbreds in 2015 and inbred checks and testers in 2016 for LS were not significant. All CVs were in a range of 11.1–47.6%, CV for LS > NLPL > DR. The significance of replications and bigger CV indicated that NCLB developed variably, which was mainly caused by three reasons: (1) Resistance of genotypes had a bigger difference from high resistance (DR < 2) to highly susceptible (DR = 7); (2) A genotype between two different genotypes might express resistance quantitatively differently; for example, a resistant genotype between two highly susceptible genotypes rating would increase by more than 1, and a susceptible genotype between two resistant genotypes rating would reduce by 1. (3) The irrigation system could have been affected by wind; one irrigation bird can reach 34.2 m in radius at mild wind environment, but it shifts to one direction when there is stronger wind, resulting in some corner rows (genotypes) not being irrigated well and developing less disease. The combined ANOVA resulted in resistance being significantly different between years. The year 2015 had a warmer May, but 2016 had a little warmer summer; therefore, the final ACHUs were almost the same, 3337.1 and 3331.6 in 2015 and 2016, respectively (Figure 1A). It had much more rainfall in June 2015 before the first inoculation, which did not affect the disease epidemic. It rained 8 days with a total of 133.4 mm from August 11 to 21 in 2016, in which 5 days rained 2.2–74.5 mm. During the same period, it rained 54.4 mm in 5 days, but it rained 35.2 and 17.8 mm on August 11 and 20, only 1.2 mm in the middle in 2015 (Figure 1B). This difference made 2016 have heavier infections than 2015.

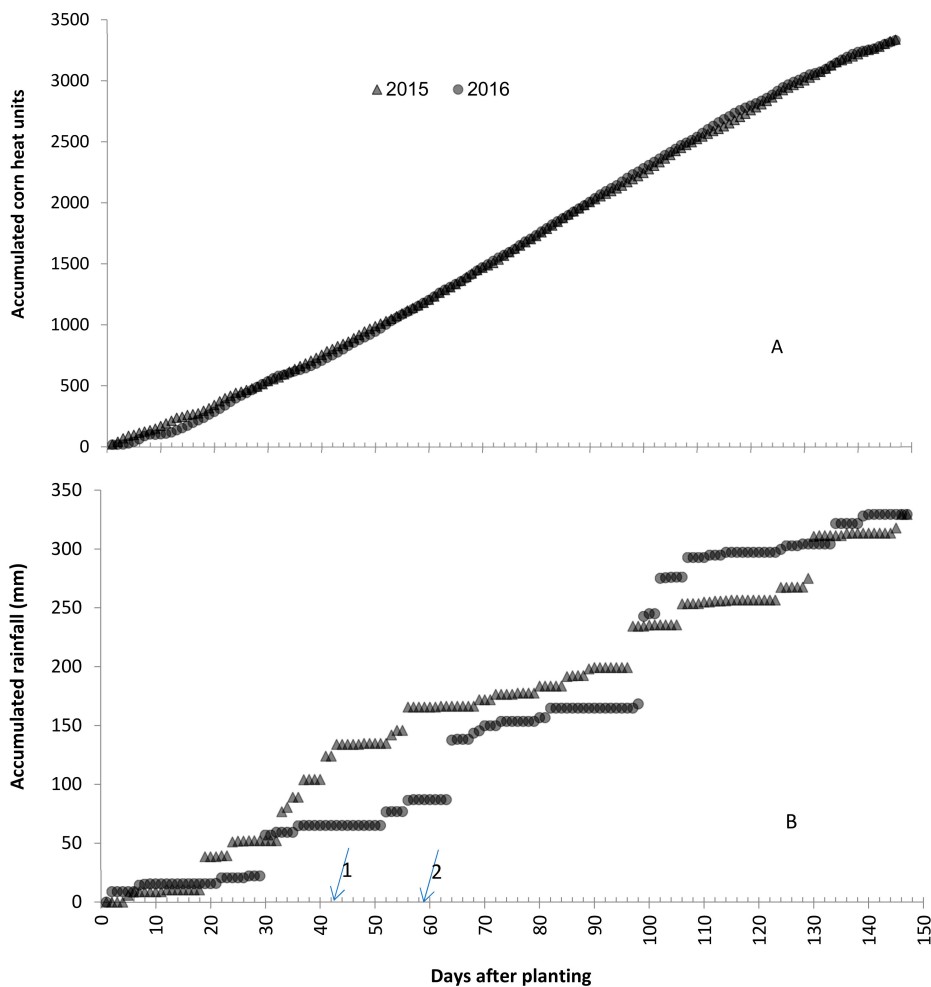

**Figure 1.** (**A**): Accumulated corn heat units (ACHU), and (**B**): accumulated rainfall (mm), from planting date (May 7th) to 30 September 2015 and 2016. Arrow 1 and arrow 2 indicate the 1st and 2nd inoculation dates. The first inoculation date was June 16th in both years; the second inoculation date was 2 July 2015 and 4 July 2016.

### 3.3. Effects of Resistant Genes on Disease Traits

Table 2 shows the detailed effects of different female and male resistant gene combinations on NCLB. If only females with resistant genes, all PG (CO428) background resistances were better than CO388 background resistances, better than no resistant gene crosses. With CO388 background, *Htm1* had less NLPL, but *Htn1* had a smaller LS. With CO428 background, PG had the lowest DR, NLPL, and LS, which meant none of the new inbreds had better resistance than CO428 in crosses. When combining all male resistant genes, similarly, with CO388 background, *Htm1* had less NLPL, but *Htn1* had a smaller LS; with CO428 background, PG*Ht1* (BLT11) had less DR and LS than PG (CO428) and others, but not statistically significant. When combining all female genes, *Htm1* (BLT03) and *Htn1* (BLT01 and BLT02) had less NLPL, but *Ht2* (A619*Ht2*) and *Ht3* (A619*Ht3*) had a smaller LS. Overall, these four male genes had a better effect than *Ht1* (A619*Ht1*) and partial resistance (T3). Table 2 also shows that when females had no resistant gene (CO388) crossed with other *Ht* genes, *Ht1*, *Ht2*, and *Ht3* had similar resistance to DR, better than partial resistance and susceptible crosses. *Ht1* (A619*Ht1*) had less NLPL and had the best overall effects of DR; *Ht2* (A619*Ht2*) had less NLPL and LS than *Ht3* (A619*Ht3*); however, Ht3 had smaller DR, which meant that CO388 × A619*Ht3* had more drying leaves caused by lesions than CO388 × A619*Ht2*. *Ht3* and partial had similar resistance to NLPL. These results indicated that if only one resistant gene is available, *Ht1* is still a good choice to make resistant crosses.

**Table 2.** Resistant gene effects on disease rating (DR), number of lesions per leaf (NLPL), and lesion size (LS, cm$^2$) of northern corn leaf blight compared with same female or male gene or combined all female or all male gene backgrounds.

| Assumed Gene from Female | Assumed Gene from Male | Count | Gene Effects and Significance | | |
|---|---|---|---|---|---|
| | | | DR | NLPL | LS (cm$^2$) |
| Effects of resistant gene(s) in the female crossed with male without resistant gene | | | | | |
| - | - | 30 | 6.5 | 8.1 | 23.8 |
| *Htm1* | - | 36 | −0.8 ** | −2.4 ** | −6.2 * |
| *Htn1* | - | 72 | −0.9 ** | −2.1 ** | −9.9 ** |
| PG | - | 12 | −1.8 ** | −4.2 ** | −11.1 ** |
| PG*HT1* | - | 12 | −1.5 ** | −3.0 ** | −10.5 ** |
| PG*Ht2* | - | 60 | −1.2 ** | −3.2 ** | −10.4 ** |
| PG*Ht3* | - | 30 | −1.5 ** | −3.7 ** | −10.8 ** |
| PG*Htm1* | - | 66 | −1.4 ** | −3.5 ** | −8.6 ** |
| PG*Htn1* | - | 60 | −1.4 ** | −3.9 ** | −8.8 ** |
| Effects of resistant gene(s) in the female after combined all male genes | | | | | |
| - | All | 54 | 5.9 | 7.0 | 19.4 |
| *Htm1* | All | 54 | −0.7 ** | −2.0 ** | −4.7 ns |
| *Htn1* | All | 108 | −0.8 ** | −1.5 * | −7.3 * |
| PG | All | 42 | −1.9 ** | −4.3 ** | −10.4 ** |
| PG*HT1* | All | 48 | −2.1 ** | −3.9 ** | −10.7 ** |
| PG*Ht2* | All | 132 | −1.6 ** | −3.5 ** | −8.9 * |
| PG*Ht3* | All | 54 | −1.6 ** | −3.4 ** | −8.9 * |
| PG*Htm1* | All | 126 | −1.6 ** | −3.7 ** | −8.3 * |
| PG*Htn1* | All | 108 | −1.5 ** | −3.6 ** | −7.3 * |
| Effects of single genes by being introduced into polygene resistance after combining all male genes | | | | | |
| PG | All | 42 | 4.0 | 2.7 | 9.0 |
| PG*HT1* | All | 48 | −0.2 ns | 0.4 ns | −0.3 ns |
| PG*Ht2* | All | 132 | 0.3 ns | 0.8 ns | 1.5 ns |
| PG*Ht3* | All | 54 | 0.3 ns | 0.9 ns | 1.6 ns |
| PG*Htm1* | All | 126 | 0.3 ns | 0.7 ns | 2.1 ns |
| PG*Htn1* | All | 108 | 0.5 ns | 0.7 ns | 3.1 ns |
| Effects of single genes in the male after combining all female genes | | | | | |
| All | - | 360 | 5.4 | 5.3 | 15.5 |
| All | *Ht1* | 72 | −1.3 ** | −1.6 ** | −6.0 ** |
| All | *Ht2* | 78 | −1.7 ** | −2.2 ** | −9.9 ** |
| All | *Ht3* | 84 | −1.8 ** | −2.6 ** | −7.8 ** |
| All | Partial | 24 | −0.3 ns | −1.0 * | −4.7 ** |
| All | *Htm1* | 42 | −1.8 ** | −3.6 ** | −6.2 ** |
| All | *Htn1* | 66 | −1.7 ** | −3.4 ** | −6.4 ** |
| Effects of single genes in the male when crossed with the same female gene | | | | | |
| - | - | 30 | 6.5 | 8.1 | 23.8 |
| - | *Ht1* | 6 | −1.7 ** | −3.4 ** | −10.8 * |
| - | *Ht2* | 6 | −1.4 * | −2.9 ** | −12.8 ** |
| - | *Ht3* | 6 | −1.5 * | −1.7 ns | −8.4 ** |
| - | Partial | 6 | −0.5 ns | −1.9 * | −5.2 ns |
| *Htm1* | - | 36 | 5.8 | 6.1 | 17.5 |
| *Htm1* | *Ht1* | 6 | −1.4 * | −2.3 ** | −10.0 ** |
| *Htm1* | *Ht2* | 6 | −1.6 * | −2.9 ** | −10.6 ** |
| *Htm1* | *Ht3* | 6 | −1.5 * | −2.4 ** | −8.5 ** |
| *Htm1* | Partial | 6 | −1.0 ns | −2.3 ns | −6.4 ns |
| *Htn1* | - | 60 | 5.7 | 6.6 | 15.1 |
| *Htn1* | *Ht1* | 12 | −1.2 ** | −1.8 ** | −4.2 * |
| *Htn1* | *Ht2* | 12 | −1.3 ** | −2.2 ** | −8.1 ** |
| *Htn1* | *Ht3* | 12 | −1.9 ** | −2.4 ** | −7.5 ** |
| *Htn1* | Partial | 12 | −0.9 ** | −3.0 ** | −7.8 ** |
| PG | - | 12 | 4.7 | 4.0 | 12.6 |
| PG | *Ht1* | 6 | −0.8 ns | −1.1 ns | −3.0 ns |
| PG | *Ht2* | 6 | −1.0 ns | −1.1 ns | −7.8 ** |
| PG | *Ht3* | 6 | −1.1 * | −2.0 * | −6.9 ** |
| PG | *Htm1* | 6 | −1.1 * | −2.5 * | −2.7 ns |
| PG | *Htn1* | 6 | −1.1 * | −2.4 * | −5.2 ns |

**Table 2.** *Cont.*

| Assumed Gene from Female | Assumed Gene from Male | Count | Gene Effects and Significance | | |
|---|---|---|---|---|---|
| | | | DR | NLPL | LS (cm$^2$) |
| PG*Ht1* | - | 12 | 5.0 | 5.1 | 13.3 |
| PG*Ht1* | *Ht1* | 6 | −1.2 * | −1.7 * | −5.4 * |
| PG*Ht1* | *Ht2* | 6 | −1.7 ** | −2.2 ** | −10.2 ** |
| PG*Ht1* | *Ht3* | 6 | −1.7 ** | −2.8 ** | −3.5 ns |
| PG*Ht1* | *Htm1* | 6 | −1.3 * | −3.2 ** | −3.7 ns |
| PG*Ht1* | *Htn1* | 12 | −1.8 ** | −3.2 ** | −6.9 ** |
| PG*Ht2* | - | 60 | 5.2 | 4.9 | 16.6 |
| PG*Ht2* | *Ht1* | 12 | −1.6 ** | −1.5 ** | −1.4 ns |
| PG*Ht2* | *Ht2* | 12 | −2.1 ** | −1.9 ** | −9.9 ** |
| PG*Ht2* | *Ht3* | 18 | −2.0 ** | −3.1 ** | −7.5 ** |
| PG*Ht2* | *Htm1* | 12 | −1.9 ** | −3.2 ** | −4.0 * |
| PG*Ht2* | *Htn1* | 18 | −1.2 ** | −3.1 ** | −3.4 * |
| PG*Ht3* | - | 30 | 5.0 | 4.4 | 13.0 |
| PG*Ht3* | *Ht1* | 6 | −1.1 * | −0.1 ns | −5.0 * |
| PG*Ht3* | *Ht2* | 6 | −2.0 ** | −2.2 ** | −8.1 ** |
| PG*Ht3* | *Ht3* | 6 | −1.4 * | −2.4 ** | −6.5 ** |
| PG*Ht3* | *Htn1* | 6 | −1.2 * | −2.6 ** | −2.5 * |
| PG*Htm1* | - | 66 | 5.1 | 4.6 | 15.2 |
| PG*Htm1* | *Ht1* | 12 | −1.4 ** | −2.1 ** | −9.3 ** |
| PG*Htm1* | *Ht2* | 12 | −1.9 ** | −2.5 ** | −9.1 ** |
| PG*Htm1* | *Ht3* | 12 | −1.6 ** | −2.7 ** | −10.1 ** |
| PG*Htm1* | *Htm1* | 12 | −1.6 ** | −2.9 ** | −6.5 ** |
| PG*Htm1* | *Htn1* | 12 | −1.7 ** | −2.8 ** | −7.4 ** |
| PG*Htn1* | - | 60 | 5.1 | 4.2 | 14.9 |
| PG*Htn1* | *Ht1* | 6 | −1.0 ns | −0.8 ns | −5.3 ** |
| PG*Htn1* | *Ht2* | 12 | −1.8 ** | −1.8 ** | −10.7 ** |
| PG*Htn1* | *Ht3* | 12 | −1.8 ** | −2.2 ** | −6.5 ** |
| PG*Htn1* | *Htm1* | 6 | −1.4 * | −2.6 ** | −5.3 ** |
| PG*Htn1* | *Htn1* | 12 | −1.1 ** | −1.8 ** | −2.9 ns |

"-" = no specific gene from female or male, "PG" = polygene resistance; Count = year # × replicate # × cross #; Gene effect = (gene mean − check mean) in the same group. "ns", "*", and "**" means significance at $p > 0.05$, $p \leq 0.05$, and $p \leq 0.01$, respectively; The row without significance markers is the group check used to compare the gene effects with its following rows.

When a female had *Htm1* (BLT03) crossed with other genes, results were similar as above, but adding the *Ht2* gene in the male had the best resistance, better than *Ht1*, *Ht3*, and partial resistance. When a female had *Htn1* (BLT01 and BLT02) crossed with other genes, adding the *Ht3* gene in the male had the best resistance, better than *Ht1*, *Ht2*, and partial resistance. *Htm1*/*Ht2* and *Htn1*/*Ht3* were two better combinations.

When a female had PG (CO428) crossed with other genes, *Ht2*, *Ht3*, *Htm1* (BLT03), and *Htn1* (BLT02) had similar resistance to DR, but *Htm1* and *Htn1* had better resistance to NLPL, while Ht2 and Ht3 had better resistance to LS. When a female had PG*Ht1* (BLT11) crossed with other genes, *Htm1* and *Htn1* expressed better resistance to NLPL, and *Ht2* expressed the best resistance to LS. PG*Ht1* × *Htn1* and PG*Ht1* × *Ht2* were two good resistant combinations. PG*Ht1* × *Ht1* had a rating of 3.7, and in such an artificial epidemic environment, the resistance was still very good. When a female had PG*Ht2* (BLT07 and BLT12) crossed with other genes, male genes *Ht2* and *Ht3* had better resistance to DR than other *Ht* genes. PG*Ht2* × *Ht2* and PG*Ht2* × *Ht3* were two better resistant combinations. When a female had PG*Ht3* (BLT13) crossed with other genes, *Ht2* expressed the best resistance to DR and LS than other *Ht* genes. PG*Ht3* × *Ht2* was the best resistant combination. When a female had PG*Htm1* (BLT05 and BLT10) crossed with other genes, the male gene *Ht2* expressed better resistance to DR than other *Ht* genes. PG*Htm1*/*Ht2* and PG*Htm1*/*Ht3* were better gene combinations. When a female had PG*Htn1* (BLT06 and BLT09) crossed with other genes, *Ht2* and *Ht3* expressed better resistance to DR than other *Ht* genes. In most cases, *Ht2* and *Ht3* had smaller LS significantly more than other genes; meanwhile, *Htm1* and *Htn1* had less NLPL while not statistically significant. PG*Htn1*/*Ht2* and PG*Htn1*/*Ht3* were better combinations.

Overall, test crosses of CO388 and its *Ht* version BLT01, BLT02, and BLT03 showed that both parents with *Ht* genes had the best resistance, followed by only one parent with *Ht* gene; no *Ht* gene with any parent was most susceptible; all significant to DR, and LS, but not all significant with NLPL. CO428 and its *Ht* version BLT05, BLT06, BLT07, BLT09, BLT10, BLT11, BLT12, and BLT13 crossed with testers showed that with the same female resistant gene background, adding male *Ht* genes increased resistance. However, with the same male *Ht* gene background, a female with a PG background had slightly better resistance than PG*Ht*(s) background, which meant that when the dominant *Ht* gene was introduced into the PG background, and PG lost one or more minor resistant gene(s). For BLT05 to BLT13, their PG backgrounds were not the same as CO428. For 10 commercial hybrid checks, 3 are with *Ht* genes, but 1 was very susceptible at the late stage and was one of the highly susceptible hybrids. It was true that *Ht1*, *Ht2*, and *Ht3* were not good enough for CO388 (SS-B73) background; only *Htm1* and *Htn1* in CO388 expressed tolerance to NCLB at the late stage. All five *Ht* genes were not good enough for CO442 background because they all died at the late stage. CO442 was one of the AAFC best GCA inbred, highly yielded with all other heterotic group inbreds, but was not a good *Ht* gene receptor.

*3.4. Line × Tester Results*

In L × T of CO388 and its *Ht* versions (Table 3), $\sigma^2_{gca}/\sigma^2_{sca} > 1$ for DR and NLPL, but not for LS, and $(\sigma^2_D/\sigma^2_A)^{1/2} < 1$ for all three traits, which meant additive effects were more important than dominant gene effects for DR and NLPL but both additive and dominant gene effects were not strong for LS. Based on genotype basis, H for DR, NLPL, and LS were 0.87, 0.82, and 0.57, respectively, but H(ns) for the four traits were 0.19, 0.18, 0.22, and 0.18, respectively. H(ns) were much smaller than H, which meant that additive gene effects were not strong, too. The GCA of lines showed that CO388 had all significant positive values for all traits; BLT01, BLT02, and BLT03 all reduced DR, but only BLT01 had significant effects; BLT01 also reduced NLPL and LS significantly; BLT02 only reduced LS, and BLT03 only reduced NLPL significantly. BLT01 and BLT02 both had *Htn1*, but their resistant sources were A553*N* and A632*HtN*, respectively. A553*N* was more resistant than A632*HtN* (Table 1); therefore, BLT01 was more resistant than BLT02. GCA for nine testers (Table 4) showed that A619*Ht1*, A619*Ht2*, A619*Ht3*, and T3 (partial resistance) all had significant negative values for all traits, implying that all resistant genes reduced NCLB development. Four genotypes without the resistant gene, A619, CL30, T1, and T2, had positive values for all four traits and increased disease development; however, CO442 reduced DR and NLPL significantly, increased LS not significantly, and expressed tolerance in the hybrid. In the SCA for 36 crosses, only 4, 4, and 6 crosses had significant positive or negative values for DR, NLPL, and LS, respectively.

**Table 3.** Genetic component, heritability (H), and heritability in the narrow sense (H(ns)), and general combining abilities of CO388-related Line × Tester analysis for disease rating (DR), number of lesions per leaf (NLPL), and lesion size (LS, cm$^2$) under artificial inoculations of northern corn leaf blight.

| Source | Resistant Gene(s) | DR | NLPL | LS (cm$^2$) |
|---|---|---|---|---|
| | | Genetic component and heritability | | |
| $\sigma^2_A$ | | 0.16 | 0.72 | 5.5 |
| $\sigma^2_D$ | | 0.04 | 0.16 | 4.6 |
| $(\sigma^2_D/\sigma^2_A)^{1/2}$ | | 2.1 | 2.13 | 1.1 |
| H | | 0.87 | 0.82 | 0.57 |
| H(ns) | | 0.19 | 0.22 | 0.18 |
| Lines | | General combining ability for lines | | |
| BLT01 | *Htn1* | −0.43 ** | −0.71 ** | −2.10 ** |
| BLT02 | *Htn1* | −0.01 ns | 0.23 ns | −2.81 ** |
| BLT03 | *Htm1* | −0.12 ns | −0.77 ** | 0.12 ns |
| CO388 | - | 0.57 ** | 1.25 ** | 4.78 ** |
| S.E(gca for lines) | | 0.07 | 0.21 | 0.77 |

**Table 3.** *Cont.*

| Source | Resistant Gene(s) | DR | NLPL | LS (cm$^2$) |
|---|---|---|---|---|
| Testers | | | General combining ability for testers | |
| A619 | - | 0.61 ** | 1.40 ** | 3.38 ** |
| A619*Ht1* | *Ht1* | −0.79 ** | −1.27 ** | −3.65 ** |
| A619*Ht2* | *Ht2* | −0.80 ** | −1.47 ** | −6.30 ** |
| A619*Ht3* | *Ht3* | −1.13 ** | −1.12 ** | −4.34 ** |
| CL30 | - | 1.30 ** | 3.55 ** | 5.36 ** |
| CO442 | - | −0.28 ** | −1.21 ** | 1.46 ns |
| T1 | - | 0.59 ** | 0.48 ns | 6.52 ** |
| T2 | - | 0.74 ** | 1.11 ** | 1.29 ns |
| T3 | Partial | −0.24 * | −1.47 ** | −3.72 ** |
| S.E(gca for testers) | | 0.10 | 0.31 | 1.16 |

"ns", "*", and "**" mean significance at $p > 0.05$, $p \leq 0.05$, and $p \leq 0.01$, respectively; "-" = no specific gene from female or male; Partial = partial resistance.

**Table 4.** Genetic component, heritability (H), and heritability in the narrow sense (H(ns)), and general combining abilities of CO428-related Line × Tester analysis for disease rating (DR), number of lesions per leaf (NLPL), and lesion size (LS, cm$^2$) under artificial inoculations of northern corn leaf blight.

| Source | Resistant Gene(s) | DR | NLPL | LS (cm$^2$) |
|---|---|---|---|---|
| Genetic component | | | | |
| $\sigma^2_A$ | | 0.08 | 0.33 | 2.09 |
| $\sigma^2_D$ | | 0.04 | 0.11 | 6.21 |
| $(\sigma^2_D/\sigma^2_A)^{1/2}$ | | 1.47 | 1.70 | 0.60 |
| H | | 0.91 | 0.93 | 0.68 |
| H(ns) | | 0.08 | 0.09 | 0.13 |
| Lines | | | | |
| BLT05 | PG*Htm1* | −0.31 ** | −0.71 ** | −1.92 ** |
| BLT06 | PG*Htn1* | 0.23 ** | 0.16 ns | 1.12 * |
| BLT07 | PG*Ht2* | −0.04 ns | 0.20 ns | −0.06 ns |
| BLT10 | PG*Htm1* | 0.02 ns | 0.18 ns | 1.00 ns |
| BLT12 | PG*Ht2* | −0.02 ns | 0.23 ns | 0.20 ns |
| CO428 | PG | 0.12 ns | −0.05 ns | −0.34 ns |
| S.E(gca for lines) | | 0.08 | 0.15 | 0.49 |
| Testers | | | | |
| A619 | - | 1.36 ** | 2.79 ** | 4.44 ** |
| A619*Ht1* | *Ht1* | −0.04 ns | 0.35 * | 0.27 ns |
| A619*Ht2* | *Ht2* | −0.63 ** | −0.15 ns | −5.15 ** |
| A619*Ht3* | *Ht3* | −0.44 ** | −0.79 ** | −3.49 ** |
| BLT02 | *Htn1* | −0.03 ns | −0.77 ** | −0.11 ns |
| BLT03 | *Htm1* | −0.36 ** | −1.10 ** | −0.09 ns |
| CO388 | - | 0.13 ns | −0.34 * | 4.13 ** |
| S.E(gca for testers) | | 0.09 | 0.16 | 0.53 |

"ns", "*", and "**" = significance at $p > 0.05$, $p \leq 0.05$, and $p \leq 0.01$, respectively; "-" = no specific gene from female or male; PG = polygene resistance.

In L × T of CO428 and its *Ht* versions (Table 4), $\sigma^2_{gca}/\sigma^2_{sca} < 1$ and $(\sigma^2_D/\sigma^2_A)^{1/2} < 1$ for DR and NLPL, which meant both additive and dominant gene effects were not strong to these traits. However, for LS, $(\sigma^2_D/\sigma^2_A)^{1/2} = 1.74$ and $\sigma^2_{gca}/\sigma^2_{sca} = 0.08$, indicating that non-additive gene action was much more important. Based on genotype basis, H for DR, NLPL, and LS were 0.91, 0.93, and 0.68, respectively, but H(ns) for the four traits were 0.08, 0.07, 0.09, and 0.13, respectively, which meant additive gene effects were also very weak. This was very different with L × T of CO388 and its *Ht* versions, which meant gene effects depended on their background. GCA of lines reduced all traits for disease development, except for BLT05, which had significant negative values. BLT06 had significant positive values for DR and LS. All GCA values for BLT07, BLT10, BLT12, and CO428 were not significant. For the GCA of testers, for DR, both A619 and CO388 had a positive value, but only A619 was significant. All five testers with *Ht* genes had negative values, which meant they reduced disease development, but three of them, *Ht2*, *Ht3*, and *Hm1*, were significant. For NLPL, A619 had a positive value, and CO388 had a negative value; adding *Ht*(S) gene

reduced lesion number, but only *Ht3*, *Htn1*, and *Htm1* reduced significantly. Of the five *Ht*(s), only *Ht2* and *Ht3* reduced LS significantly. There were 4, 5, and 9 of 42 crosses that had significant SCA. Both L × T results showed that SCA effects were less important than GCA, but gene effects depend on genotypes.

### 3.5. MPH and BPH Results

Table 5 shows that MPH and BPH had similar positive or negative for DR and NLPL in most crosses; however, most MPH and BPH were positive for LS, with some BPH over 100% up to 566.3%. NCLB lesion is limited by the veins of the leaf, resulting in smaller leaves having smaller lesions. In CO388-related L × T, tester CL30 and T1, and in CO428-related L × T, line BLT06 had relatively smaller leaves, and their LS(s) were smaller, too. However, when crossed with normal size CO388, BLT01, BLT02, and BLT03, their F1 plants were normal; therefore, their LS(s) were normal, too. It seemed that MPH and BPH for LS were highly related to the heterosis of the leaf size. LS is not a good trait for MPH and BPH.

**Table 5.** The mid-parent heterosis (MPH) and the better-parent heterobeltiosis (BPH) of 36 CO388-related and 42 CO428-related Line × Tester crosses for disease rating (DR), number of lesions per leaf (NLPL), and lesion size (LS, cm$^2$) under artificial inoculations of northern corn leaf blight.

| Genotype | Resistant Gene(S) | DR | | NLPL | | LS (cm$^2$) | |
|---|---|---|---|---|---|---|---|
| | | MPH | BPH | MPH | BPH | MPH | BPH |
| CO388 × A619 | -/- | −4.5 ns | −3.8 ns | −2.0 ns | −1.6 ns | 23.8 ns | 28.6 ns |
| CO388 × A619*Ht1* | -/*Ht1* | −16.2 ** | −3.7 ns | −32.9 ** | −19.4 ns | 4.0 ns | 22.7 ns |
| CO388 × A619*Ht2* | -/*Ht2* | −9.1 ns | 8.4 ns | −20.2 ns | 6.8 ns | 1.1 ns | 49.2 ns |
| CO388 × A619*Ht3* | -/*Ht3* | −13.8 ** | −0.2 ns | −7.6 ns | 10.9 ns | 36.7 ns | 83.8 * |
| CO388 × CL30 | -/- | 2.6 ns | 6.5 ns | 38.1 ** | 50.7 ** | 73.0 ** | 96.2 ** |
| CO388 × CO442 | -/- | −9.9 * | −6.6 ns | −26.5 * | −24.9 * | 60.4 * | 65.9 ** |
| CO388 × T1 | -/- | −4.7 ns | −1.0 ns | −9.5 ns | −6.5 ns | 77.6 ** | 91.5 ** |
| CO388 × T2 | -/- | 0.7 ns | 3.3 ns | 17.3 ns | 26.4 * | 57.2 * | 62.5 ** |
| CO388 × T3 | -/Partial | −4.5 ns | −1.4 ns | −11.0 ns | 7.5 ns | 54.0 ns | 132.6 ** |
| BLT01 × A619 | *Htn1*/- | −6.8 ns | 0.8 ns | −17.0 ns | −13.1 ns | 67.5 * | 215.3 ** |
| BLT01 × A619*Ht1* | *Htn1*/*Ht1* | −22.0 ** | −16.9 ** | −38.0 ** | −29.7 * | 26.6 ns | 82.9 ns |
| BLT01 × A619*Ht2* | *Htn1*/*Ht2* | −20.2 ** | −11.9 * | −34.2 * | −17.1 ns | 4.0 ns | 20.4 ns |
| BLT01 × A619*Ht3* | *Htn1*/*Ht3* | −34.4 ** | −29.6 ** | −48.1 ** | −41.1 ** | −36.6 ns | −20.1 ns |
| BLT01 × CL30 | *Htn1*/- | 3.5 ns | 15.7 ** | 16.1 ns | 20.3 ns | 23.7 ns | 163.2 ** |
| BLT01 × CO442 | *Htn1*/- | −31.4 ** | −23.6 ** | −47.3 ** | −43.2 ** | 45.8 ns | 148.0 * |
| BLT01 × T1 | *Htn1*/- | −12.3 ** | −1.9 ns | −15.9 ns | −14.4 ns | 122.0 ** | 340.6 ** |
| BLT01 × T2 | *Htn1*/- | −6.2 ns | 3.5 ns | −20.7 ns | −18.9 ns | 23.7 ns | 131.5 * |
| BLT01 × T3 | *Htn1*/Partial | −24.3 ** | −21.4 ** | −55.9 ** | −49.7 ** | 23.4 ns | 40.2 ns |
| BLT02 × A619 | *Htn1*/- | 3.0 ns | 14.0 ** | 18.6 ns | 43.0 ** | 84.6 * | 283.5 ** |
| BLT02 × A619*Ht1* | *Htn1*/*Ht1* | −6.8 ns | −2.9 ns | −6.6 ns | −5.8 ns | 56.3 ns | 146.6 * |
| BLT02 × A619*Ht2* | *Htn1*/*Ht2* | −5.7 ns | 1.8 ns | −11.3 ns | −3.4 ns | 21.7 ns | 52.2 ns |
| BLT02 × A619*Ht3* | *Htn1*/*Ht3* | −20.0 ** | −16.1 ** | −14.7 ns | −14.0 ns | 62.8 ns | 122.6 ns |
| BLT02 × CL30 | *Htn1*/- | 2.7 ns | 17.7 ** | 33.0 * | 46.5 ** | 10.4 ns | 160.2 * |
| BLT02 × CO442 | *Htn1*/- | −16.4 ** | −4.5 ns | −25.2 * | −7.0 ns | 46.6 ns | 174.0 * |
| BLT02 × T1 | *Htn1*/- | −6.8 ns | 6.8 ns | −6.5 ns | 9.3 ns | 21.3 ns | 166.1 * |
| BLT02 × T2 | *Htn1*/- | −7.8 ns | 4.2 ns | 0.5 ns | 12.2 ns | 8.2 ns | 123.3 ns |
| BLT02 × T3 | *Htn1*/Partial | −10.9 * | −5.4 ns | −25.2 ns | −25.0 ns | 19.0 ns | 45.7 ns |
| BLT03 × A619 | *Htm1*/- | −6.9 ns | 1.1 ns | −13.1 ns | 3.4 ns | 73.3 * | 265.7 ** |
| BLT03 × A619*Ht1* | *Htm1*/*Ht1* | −17.2 ** | −12.1 * | −35.6 * | −35.4 * | 12.9 ns | 80.7 ns |
| BLT03 × A619*Ht2* | *Htm1*/*Ht2* | −18.1 ** | −10.0 ns | −39.8 * | −33.6 ns | 32.1 ns | 67.2 ns |
| BLT03 × A619*Ht3* | *Htm1*/*Ht3* | −17.9 ** | −12.3 * | −36.2 * | −36.0 * | 52.2 ns | 110.9 ns |
| BLT03 × CL30 | *Htm1*/- | 7.0 ns | 20.1 ** | 59.3 ** | 73.3 ** | 116.2 ** | 418.1 ** |
| BLT03 × CO442 | *Htm1*/- | −25.3 ** | −16.4 ** | −64.4 ** | −56.3 ** | 66.0 ns | 215.1 ** |
| BLT03 × T1 | *Htm1*/- | −5.5 ns | 6.1 ns | −22.7 ns | −10.8 ns | 87.3 ** | 317.6 ** |
| BLT03 × T2 | *Htm1*/- | −1.3 ns | 9.4 ns | −1.0 ns | 9.1 ns | 66.5 * | 249.1 ** |
| BLT03 × T3 | *Htm1*/Partial | −17.3 ** | −13.8 ** | −34.7 * | −34.1 * | 103.9 ns | 152.9 * |

**Table 5.** *Cont.*

| Genotype | Resistant Gene(S) | DR | | NLPL | | LS (cm$^2$) | |
|---|---|---|---|---|---|---|---|
| | | MPH | BPH | MPH | BPH | MPH | BPH |
| CO428 × A619 | PG/- | 9.8 ns | 73.2 ** | 25.3 * | 293.5 ** | 34.4 ns | 189.4 ** |
| CO428 × A619*Ht1* | PG/*Ht1* | −2.0 ns | 29.3 ** | −21.3 ns | 89.1 * | 29.1 ns | 110.3 ** |
| CO428 × A619*Ht2* | PG/*Ht2* | −4.2 ns | 21.4 * | −9.4 ns | 89.1 * | −19.3 ns | 3.8 ns |
| CO428 × A619*Ht3* | PG/*Ht3* | −9.4 ns | 18.6 ns | −45.7 ** | 30.4 ns | −12.3 ns | 23.6 ns |
| CO428 × BLT02 | PG/*Htn1* | −15.0 * | 18.0 ns | −56.0 ** | 4.3 ns | 53.7 ns | 57.4 ns |
| CO428 × BLT03 | PG/*Htm1* | −17.0 * | 17.8 ns | −60.4 ** | −4.3 ns | 112.5 ** | 115.2 ** |
| CO428 × CO388 | PG/- | −14.2 * | 34.1 ** | −60.8 ** | 23.9 ns | 30.6 ns | 165.8 ** |
| BLT05 × A619 | PG*Htm1*/- | −13.7 * | 12.6 ns | −23.2 * | 96.6 ** | −11.5 ns | 25.1 ns |
| BLT05 × A619*Ht1* | PG*Htm1*/*Ht1* | −15.8 * | −6.4 ns | −39.3 ** | 20.3 ns | −46.8 * | −40.2 ns |
| BLT05 × A619*Ht2* | PG*Htm1*/*Ht2* | −32.6 ** | −27.6 ** | −49.3 ** | −11.9 ns | −35.9 ns | −31.1 ns |
| BLT05 × A619*Ht3* | PG*Htm1*/*Ht3* | −30.5 ** | −23.2 ** | −64.1 ** | −28.8 ns | −64.7 ** | −64.6 ** |
| BLT05 × BLT02 | PG*Htm1*/*Htn1* | −28.2 ** | −16.4 * | −61.0 ** | −23.7sn | 25.6 ns | 71.2 ns |
| BLT05 × BLT03 | PG*Htm1*/*Htm1* | −35.5 ** | −23.4 ** | −75.3 ** | −50.8 ns | 0.9 ns | 36.9 ns |
| BLT05 × CO388 | PG*Htm1*/- | −28.2 ** | −7.0 ns | −53.3 ** | 20.3 ns | 22.9 ns | 65.8 ** |
| BLT06 × A619 | PG*Htn1*/- | 13.3 * | 91.5 ** | 8.4 ns | 365.6 ** | 44.9 * | 361.6 ** |
| BLT06 × A619*Ht1* | PG*Htn1*/*Ht1* | 6.4 ns | 49.3 ** | −1.4 ns | 218.8 ** | 44.4 ns | 237.0 ** |
| BLT06 × A619*Ht2* | PG*Htn1*/*Ht2* | −10.6 ns | 20.4 ns | −7.9 ns | 156.3 ** | −20.4 ns | 41.5 ns |
| BLT06 × A619*Ht3* | PG*Htn1*/*Ht3* | −10.7 ns | 24.4 * | −32.4 * | 118.8 * | 52.1 ns | 200.6 ** |
| BLT06 × BLT02 | PG*Htn1*/*Htn1* | 3.8 ns | 53.6 ** | −20.6 ns | 153.1 ** | 146.6 ** | 233.2 ** |
| BLT06 × BLT03 | PG*Htn1*/*Hm1* | −10.6 ns | 35.3 ** | −52.9 ** | 53.1 ns | 155.3 ** | 240.4 ** |
| BLT06 × CO388 | PG*Htn1*/- | −4.5 ns | 59.8 ** | −45.1 ** | 137.5 ** | 122.6 ** | 566.3 ** |
| BLT07 × A619 | PG*Ht2*/- | 8.6 ns | 67.9 ** | 12.1 ns | 175.8 ** | 42.3 * | 130.2 ** |
| BLT07 × A619*Ht1* | PG*Ht2*/*Ht1* | −1.5 ns | 27.6 ** | −6.3 ns | 79.0 ** | 67.4 ** | 111.5 ** |
| BLT07 × A619*Ht2* | PG*Ht2*/*Ht2* | −17.1 * | 3.2 ns | −19.2 ns | 35.5 ns | −50.4 ns | −48.8 ns |
| BLT07 × A619*Ht3* | PG*Ht2*/*Ht3* | −18.1 * | 5.3 ns | −52.7 ** | −9.7 ns | −14.2 ns | −4.4 ns |
| BLT07 × BLT02 | PG*Ht2*/*Htn1* | −14.9 * | 15.8 ns | −41.9 ** | 9.7 ns | 64.1 * | 97.9 * |
| BLT07 × BLT03 | PG*Ht2*/*Htm1* | −23.4 ** | 6.6 ns | −53.8 ** | −11.3 ns | 88.3 ** | 129.9 ** |
| BLT07 × CO388 | PG*Ht2*/- | −19.2 ** | 23.9 * | −53.1 ** | 16.1 ns | −58.2 ** | −35.7 ns |
| BLT10 × A619 | PG*Htm1*/- | 12.3 ns | 88.4 ** | 19.9 ns | 462.1 ** | 47.1 * | 205.9 ** |
| BLT10 × A619*Ht1* | PG*Htm1*/*Ht1* | −5.7 ns | 31.5 ** | −12.7 ns | 206.9 ** | 23.3 ns | 94.8 * |
| BLT10 × A619*Ht2* | PG*Htm1*/*Ht2* | −13.4 ns | 15.8 ns | −28.0 ns | 117.2 * | −24.3 ns | −5.2 ns |
| BLT10 × A619*Ht3* | PG*Htm1*/*Ht3* | −4.0 ns | 6.6 ns | −28.4 ns | 151.7 ** | 5.0 ns | 43.8 ns |
| BLT10 × BLT02 | PG*Htm1*/*Htn1* | −18.3 * | 31.5 ** | −40.3 * | 106.9 ns | 41.6 ns | 41.8 ns |
| BLT10 × BLT03 | PG*Htm1*/*Htm1* | −8.0 ns | 15.8 ns | −30.7 ns | 144.8 * | 126.9 ** | 129.1 ** |
| BLT10 × CO388 | PG*Htm1*/- | −12.8 ns | 44.9 ** | −31.4 ** | 224.1 ** | 100.5 ** | 294.3 ** |
| BLT12 × A619 | PG*Ht2*/- | 18.2 ** | 32.9 ** | 50.9 ** | 411.9 ** | 39.1 * | 125.5 ** |
| BLT12 × A619*Ht1* | PG*Ht2*/*Ht1* | −15.4 ns | 44.9 ** | −11.5 ns | 128.6 ** | 11.5 ns | 41.0 ns |
| BLT12 × A619*Ht2* | PG*Ht2*/*Ht2* | −23.0 ** | 20.1 ns | 5.3 ns | 135.7 ** | −53.0 * | −51.5 ns |
| BLT12 × A619*Ht3* | PG*Ht2*/*Ht3* | −15.7 * | 38.4 ** | −53.9 ** | 19.0 ns | −41.4 ns | −34.5 ns |
| BLT12 × BLT02 | PG*Ht2*/*Htn1* | 13.0 ns | 62.1 ** | −50.5 ** | 26.2 ns | 140.3 ** | 189.6 * |
| BLT12 × BLT03 | PG*Ht2*/*Htm1* | −21.4 ** | 15.4 ns | −56.0 ** | 14.3 ns | 33.8 ns | 63.1 ns |
| BLT12 × CO388 | PG*Ht2*/- | −21.7 ** | 26.9 * | −56.8 ** | 47.6 ns | 18.6 ns | 82.7 ** |

"ns", "*", and "**" means significance at $p > 0.05$, $p \leq 0.05$, and $p \leq 0.01$, respectively; "-" = no specific gene from female or male, Partial = partial resistance, PG = polygene resistance.

For resistance, negative MPH and BPH were better than positive MPH and BPH; the smaller, the better, which means that the F1 hybrid had resistance close to or better than the more resistant parent, and dominant gene action was present. In CO388-related L × T of Table 5 for DR, line BLT01 had the smallest MPH, followed by BLT03 and BLT02; CO388 had the biggest MPH; for NLPL, line BLT01 had the smallest MPH, followed by BLT03 and CO388; BLT02 had the biggest MPH. It proved that both *Htn1* and *Htm1* genes increased resistance. In CO388-related L × T, testers with *Ht1*, *Ht2*, *Ht3*, and partial resistance all had negative MDP, 13 out of 16 MDPs were significant, implying the presence of dominant gene action. Some BPH(s) were significantly negative in BLT01, BLT02, and BLT03 for DR and NLPL, implying that the presence of over-dominant gene action also existed, but not often. Testers without resistant genes responded differently for DR and NLPL; CL30 had all positive values; CO442 had all negative values, and A619, T1, and T2 had both positive and negative values for DR and NLPL. As a male, CO442 expressed significant tolerance to DR and NLPL.

In CO428-related L × T of Table 5 for DR, line BLT05 had the smallest MPH, followed by BLT07, BLT12, BLT10, and CO428; BLT06 had the biggest MPH. For NLPL, line BLT05

had the smallest MPH, followed by CO428, BLT07, BLT12, and BLT06; BLT10 had the biggest MPH. Overall, BLT05 had the best resistance, and BLT06 had the least resistance, which was similar to the GCA results. For testers, A619 crossed with CO428, BLT06, BLT07, BLT10, and BLT12 had positive MPH for DR and NLPL; meanwhile, CO388 crossed with them had significant negative values to these two traits; it proved that CO388 expressed tolerance in hybrids. For others that were crossed with five testers with *Ht* genes, except BLT06 × A619*Ht1* and BLT06 × BLT02, their MPH all were negative values, meaning dominant gene action was present. Only BLT05 crossed with five *Ht* gene lines (A619*Ht1*, A619*Ht2*, A619*Ht3*, BLT02, and BLT03) showed negative BPH to DR; some were significant, implying that over-dominant gene action existed.

*3.6. Discussion*

This study again proved the NCLB disease cycle difference between susceptible and resistant genotypes [1–5], especially the difference in the latent period. (1) The ground diseased leaf powder needed about 24 h in a wet whorl to produce spores for invasion. (2) This study had complicated genotypes with many resistant gene combinations and a complicated disease population with many races [54], which developed different types of lesions. After the first inoculation, S lesions showed typical symptoms and sporulation approximately in 10–14 days. MS and MR lesions showed typical symptoms and sporulation in approximately 14–28 days. HR and R lesions from some crosses needed approximately 28 days to show typical symptoms and had sporulation at the late stage. Two commercial hybrids never had sporulation. (3) After the second inoculation, the latent period overlapped. (4) When plants got older, the latent period got shorter, but sporulation got smaller, too.

As discussed earlier, NCLB development was affected by genotypes, pathogen pathogenicity, and climatic and irrigation conditions. NCLB epidemic was also affected by temporal- [42] and spatial- [73] systems. In this study, 130 crosses and hybrid checks had silking days of 66–89 days, on average 77 days, and had a plant height of 183–326 cm, on average 254 cm. For early mature genotypes, their plants developed earlier and suffered longer under disease epidemic environments than later maturity genotypes. The concentrations of spores were different in different spatial heights; the top parts of taller genotypes suffered less concentration than lower parts and shorter genotypes. These might be reasons why more tolerant genotypes were screened from later and taller genotypes; even with the same resistant genes, taller hybrids were more resistant than shorter inbreds. The area under the disease progress curve and apparent infection rate [8,19] have some functions of the temporal system. More spatial- and temporal-system studies are needed for disease development.

NCLB caused dead leaf areas by diseased lesions and drying leaf edges or tips. In this study, ear-leaf length, ear-leaf width, and ear-leaf area ranged from 42–104 cm, 6–12 cm, and 222–843 cm$^2$, on average 79 cm, 9.4 cm, and 565 cm$^2$, respectively. Few lesions on smaller leaf plants caused a bigger percentage of diseased leaves. All CL30 crosses were susceptible, DR > 5.5, because they are shorter and smaller than others. CO428 family × CO388 family, CO388 family × CO442, and CO4288 family × CO442 all had long and wide leaves, which showed tolerance to NCLB, though CO442 was highly susceptible. This kind of tolerance was caused by plant architecture and could not be explained by additive or dominant gene effects.

Average gene effects can be expressed by percentage instead of real data. When resistant genes were in females, *Htm1*, *Htn1*, and PG effects (%) to DR were −13.9, −15.3, and −18.2, to NLPL were −29.4, −32.3, and −38.2, to LS were −26.5, −42.0, and −44.5, respectively. When resistant genes were in males, *Ht1*, *Ht2*, *Ht3*, *Htm1*, *Htn1*, and partial effects (%) to DR were −24.8, −28.5, −29.9, −12.6, −8.6, and −7.2, to NLPL were −42.0, −45.6, −47.6, −29.5, −20.7, and −23.6, to LS were −32.8, −57.8, −43.7, −23.8, −30.9, and −31.3, respectively. Using these percentage data to make MPH and BPH comparisons, it may be possible to resolve the problem in Table 5.

This study included five *Ht genes*, PG, PG*Ht*(s), and their combinations, and its results can be used for better selections. If only a single dominant gene is available, *Ht1* is still a good choice. However, if the pathogen population can overcome *Ht1*, other genes should be selected. CO428, with greener leaves and less sensitive reactions to NCLB, common rust, eyespot, and other leaf diseases, is a better PG choice than its resistant source H99. Compared to single dominant genes, PG and PG*Ht*(s) had better resistance to DR, NLPL, and LS.

## 4. Conclusions

After 10 years of resistant breeding, *Htm1* and *Htn1* were successfully introduced into a high-yield GCA inbred CO388, and *Ht1*, *Ht2*, *Ht3*, *Htm1*, and *Htn1* were successfully introduced into a polygene-resistant inbred CO428. If only one parent was with resistance, PG had the best resistance, followed by *Ht1*, which meant if no PG parent was available, the Ht1 parent was still the best choice. The resistance of PG*Ht*(s) was not always better than PG, meaning that one or more minor resistant gene(s) might have been lost when *Ht*(s) was introduced into PG. If both parents were with resistance, *Ht2* and *Ht3* reduced more LS, and *Htm1* and *Htn1* reduced more NLPL. PG and its *Ht* version crossed with *Ht2*, *Ht3*, *Htm1*, and *Htn1* had good resistance effects. In this study, *Htm1/Ht2* and *Htn1/Ht3*, PG/*Htm1*, PG/*Htn1*, PG*Ht1/Htn1*, PG*Ht1/Ht2*, PG*Ht1/Ht3*, PG*Ht2/Ht2*, PG*Ht2/Ht3*, PG*Ht3/Ht2*, PG*Htm1/Ht2*, PG*Htn1/Ht2*, and PG*Htn1/Ht3* were better combinations. Two L × T analyses showed that GCA effects were bigger than SCA effects for DR and NLPL in CO388-related L × T but inverse in CO428-related L × T; only strong dominant effects were found for LS in CO428-related L × T. Heritability based on genotype base was high, but heritability in the narrow sense was low in both L × T, which meant that these resistances were heritable, but additive genetic effects were low. Line BLT01 had the best GCA in CO388-related L × T, and line BLT05 had the best GCA in CO428-related L × T. MPH and BPH results showed that dominant and over-dominant gene action existed for DR and NLPL. LS was not a good trait for MPH and BPH analysis because it was affected by the leaf size. PG and PG*Ht*(s) are not only good choices for NCLB resistance breeding, reducing risks of different races of the pathogen, but also good choices for multiple-resistance breeding because PG is resistant to multiple leaf diseases.

**Author Contributions:** Conceptualization, X.Z. and L.M.R.; methodology, X.Z. and L.M.R.; software, X.Z.; validation, X.Z. and Aida Kebede; formal analysis, X.Z.; investigation, X.Z., T.W., K.K.J. and J.W.; data curation, X.Z.; writing—original draft preparation, X.Z.; writing—review and editing, A.K. and X.Z.; supervision, L.M.R. and Aida Kebede; project administration, A.K.; funding acquisition, L.M.R. All authors have read and agreed to the published version of the manuscript.

**Funding:** This project was supported by the Agriculture and Agri-Food Canada Growing Forward Partnership with the Canadian Field Crop Research Alliance and Grain Farmers of Ontario, which obtained funding through "Growing Forward 2" (GF2), a federal-provincial-territorial initiative. The Agricultural Adaptation Council assisted in the delivery of GF2 in Ontario.

**Data Availability Statement:** Data available by contacting correspondence author.

**Acknowledgments:** We Thank USDA-ARS and Cornell University, USA, for providing resistant sources; MBS Genetics, L.L.C., and Thurston Genetics for providing us tester seeds; and AgReliant Genetics Inc., Westfield, Indiana; Maizex Seeds Inc., Tilbury, ON, Canada; Pioneer Hi-Bred Ltd., Des Moines, IA, USA; and Deklab (Monsanto Canada Inc., Ottawa, ON, Canada) for providing check hybrid seeds. We also thank international students Shiyuan Lu and Qing Zhao from Yangzhou University, China, for field work and data collection. Thank Reza Jamaly for helping work on references with EndNote.

**Conflicts of Interest:** The authors declare no conflict of interest.

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
