# Peer review of "Resistance Breeding for Northern Corn Leaf Blight with Dominant Genes, Polygene, and Their Combinations—Effects on Disease Traits"

_agronomy, doi:10.3390/agronomy13041096_

Round 1
Reviewer 1 Report
agronomy-2303917-peer-review-v1
The paper: Resistance Breeding to Northern Corn Leaf Blight with Dominant Genes, Polygene, and Their Combinations - Effects to Disease Traits.
I would like to thank the authors for this outstanding work. The manuscript idea is great, the parts of the manuscript were narrated in a characteristic and interesting way.
Title: the title of the manuscript is appropriate.
Abstract: good section
Introduction:
The paragraph from line 40 to 45 indicates that there are previous studies that are completely similar to the current study. Please highlight the difference or the novelty.
Materials and Methods:
The materials and methods have been written extensively in a well-organized manner.
Results
The results pane has been displayed in a smooth and simplified manner.
Discussion
The discussion part is not commensurate with the effort made in the study. I suggest improving it with more recent references.
Conclusions: Excellent.
Best Regards
Author Response
Reviewer 1 is a good reviewer, give good suggestions.
Thanks.

Reviewer 2 Report
Dear Dr Chief in editor, editors, authors.
Happy day
The paper reflect excessive work but need re-organization in some parts.
1- How can you confirm that the crossover is successful. You did not describe any of the molecular tools that could confirm that the crossing over has been happened correctly never describe in details how you mad such crossing. Kindly add more details.
2- What you mean exactly with this sentence. Kindly give more details. And what is happened after the 20 days. line 33-34
"Thus, one complete disease cycle on susceptible plants takes place within 14 days, whereas it takes about20 days on resistant plants"
3- Kindly explain in more details what did you benefit from running the statistical analysis.
4- Kindly link the observed data to the statistical analysis ones.
with my pleasure
5- Kindly, look if you can update the references.
Amro Amara

Author Response
Reviewer 2 is a good reviewer, very polite and give detail comments.
Thanks.
